# Bright single photon emitters with enhanced quantum efficiency in a two-dimensional semiconductor coupled with dielectric nano-antennas

Luca Sortino [1,2 ✉], Panaiot G. Zotev[1], Catherine L. Phillips [1], Alistair J. Brash [1], Javier Cambiasso[3], Elena Marensi[4], A. Mark Fox[1], Stefan A. Maier [2,3], Riccardo Sapienza [3] & Alexander I. Tartakovskii [1 ✉]

Single photon emitters in atomically-thin semiconductors can be deterministically positioned using strain induced by underlying nano-structures. Here, we couple monolayer $WSe_2$ to high-refractive-index gallium phosphide dielectric nano-antennas providing both optical enhancement and monolayer deformation. For single photon emitters formed on such nano-antennas, we find very low (femto-Joule) saturation pulse energies and up to $10^4$ times brighter photoluminescence than in $WSe_2$ placed on low-refractive-index $SiO_2$ pillars. We show that the key to these observations is the increase on average by a factor of 5 of the quantum efficiency of the emitters coupled to the nano-antennas. This further allows us to gain new insights into their photoluminescence dynamics, revealing the roles of the dark exciton reservoir and Auger processes. We also find that the coherence time of such emitters is limited by intrinsic dephasing processes. Our work establishes dielectric nano-antennas as a platform for high-efficiency quantum light generation in monolayer semiconductors.

[1] Department of Physics and Astronomy, University of Sheffield, Sheffield S3 7RH, UK. [2] Chair in Hybrid Nanosystems, Nanoinstitute Munich, Faculty of Physics, Ludwig-Maximilians-Universität München, 80539 München, Germany. [3] The Blackett Laboratory, Department of Physics, Imperial College London, London SW7 2BW, UK. [4] IST Austria, Am Campus 1, 3400 Klosterneuburg, Austria. ✉email: luca.sortino@physik.uni-muenchen.de; a.tartakovskii@sheffield.ac.uk

Single-photon emitters (SPEs) in two-dimensional (2D) semiconducting WSe$_2$[1–5] open attractive perspectives for few-atom-thick devices for quantum technologies owing to favorable excitonic properties[6] and the integration with arbitrary substrates, including nanostructured surfaces[7,8]. Several theoretical models have been proposed to provide insight into the origin of SPEs observed in the cryogenic photoluminescence (PL) spectra of 2D WSe$_2$[9–12]. Their occurrence was explained by the presence of strain-induced potential traps for excitons[9], momentum-dark states[10], or various types of defects[11,12]. While the exact origin is still under debate, the first significant steps have been made to integrate WSe$_2$ SPEs in devices, including electroluminescent structures[13–15], waveguides[16,17], and tunable high-Q microcavities[18].

An appealing approach for the scalable and controllable fabrication of SPEs in WSe$_2$ is the use of strain engineering for their deterministic positioning. Based on this idea, SiO$_2$[19] or polymer nanopillars[20] have been employed to induce arrays of SPEs in atomically thin WSe$_2$. In a similar approach, nanostructures made of noble metals were also employed where, due to the enhancement of the near-field intensity by plasmonic resonances, increased spontaneous emission rates were demonstrated[21]. However, plasmonic nanoantennas are known for large nonradiative losses, particularly detrimental for quantum-technology applications. Thus, special care needs to be taken to separate SPEs from metallic surfaces with a dielectric spacer, which on the other hand will reduce the desired near-field coupling[21].

In contrast, the high-refractive-index dielectric materials used in our work offer a lossless alternative to metals[22]. Sub-wavelength dielectric nano-antennas exhibit optical Mie resonances carrying both electric and magnetic responses[23]. High-index dielectric nano-antennas have also been recently shown to provide an efficient approach for the enhancement of light–matter interaction as well as improved emitted light directionality in molecules[24], colloidal quantum dots[25], and excitons in 2D semiconductors[7,26].

Here, we realize SPEs by placing monolayer WSe$_2$ on top of dielectric nano-antennas made from high-refractive index GaP. Such SPEs show considerably enhanced PL counts per unit excitation power compared with previously reported emitters in WSe$_2$ as well as the SPEs realized on low-index SiO$_2$ nanopillars in our work. The nano-antennas act as broadband optical cavities and also create strain pockets where the SPEs form. For such SPEs, our numerical simulations predict PL enhancement factors $\langle EF \rangle$[7,27] up to 800, compared with a more standard realization of WSe$_2$ SPEs on SiO$_2$ pillars[19]. However, owing to the substantially enhanced quantum efficiency (QE) in the SPEs on GaP nanoantennas, we can employ low-excitation powers below fJ per laser pulse for their efficient operation. Thus, when we compare SPEs on GaP nanoantennas and SiO$_2$ pillars experimentally, we find up to 10$^4$ brighter PL per unit laser power in the former system. We show that this substantial improvement in the operation of the SPEs is related to the low QE of the SPEs on SiO$_2$, $4 \pm 2\%$ on average, compared with $21 \pm 3\%$ average and up to 86% maximum in SPEs on GaP nanoantennas. For the latter, we find that for the pumping laser repetition rate of 80 MHz, the SPE generates an effective single photons rate as high as 69 MHz under laser pulse energies around a fJ, corresponding to the single-photon rate of 5.5 MHz at the first lens.

Our approach allows further insight into the exciton dynamics in the hybrid 2D/0D system (2D monolayer/SPE) at very low-excitation densities. We observe that as the pumping power is increased, exciton–exciton annihilation[28,29], in our case of dark excitons, prevents the efficient population of SPEs. This insight allows to develop a fuller understanding of the limitations of the low QE systems, in our case the SPEs on SiO$_2$. There, the requirement for increased pumping power leads to a fast nonradiative depletion of excitons in 2D WSe$_2$ and eventually results in a low single-photon generation rate, that cannot be overcome by further increasing the pumping power. Our results thus highlight that the high-refractive-index nanoantennas, exhibiting near-field optical enhancement, provide strong advantages for producing bright SPEs in WSe$_2$ monolayers.

## Results

**Optical properties of WSe$_2$ SPEs positioned on GaP nanoantennas.** We use GaP nanoantennas composed of two closely spaced nanopillars (Fig. 1a), referred to as "dimer" below (see also Supplementary Fig. 1a). They exhibit an enhancement of the electromagnetic near-field intensity, as a result of the high refractive index ($n_{GaP} \approx 3.2$) and the hybridization of the optical resonances of each individual pillars (see Supplementary Notes I and II). This is demonstrated in Fig. 1a, where we show the calculated electric field intensity of the scattered radiation, $|E|^2$, normalized by the intensity $|E_0|^2$ of the normally incident plane wave with linear polarization along the axis connecting the centers of the nanopillars. The profile in Fig. 1a corresponds to the top surface of the GaP dimer ($z = 200$ nm) having individual pillar radii of 150 nm. The enhancement of $|E|^2$ compared with $|E_0|^2$ exceeds ten times and is particularly pronounced in the gap between the pillars[7,24,30]. The field is also strongly enhanced at the outer edges of the dimer where we expect SPEs to be located, as discussed further below and in Supplementary Note I. As shown in Fig. 1b, under the same excitation conditions, a SiO$_2$ nanopillar ($r = 150$ nm, $h = 100$ nm) does not show strong electromagnetic resonances, as a consequence of its low refractive index ($n_{SiO_2} \approx 1.5$).

A dipole emitter, such as an exciton in an SPE, spectrally and spatially overlapping with the near-field of the antenna, is expected to exhibit an enhanced light emission intensity[27]. This is a result of the product of the three main factors giving rise to the PL enhancement factor $\langle EF \rangle$[7,27]. Depending on the relative position and orientation of its dipole moment, the emitter experiences an increased local density of states and thus an enhanced spontaneous emission rate[31] via the Purcell effect, introducing a factor $F_p$, directly improving the overall quantum efficiency, $QE = F_p \gamma_{rad}/(F_p \gamma_{rad} + \gamma_{nr})$, where $\gamma_{rad}$ and $\gamma_{nr}$ are the rates of the radiative and nonradiative decay, respectively. In the absence of the Purcell effect, $F_p = 1$ and we assumed $\gamma_{rad} \ll \gamma_{nr}$. The antenna also modifies the dipole far-field emission pattern, leading to an increased light collection efficiency above the antenna ($\eta_{obj}$) within the given numerical aperture (NA) of the objective lens used in the detection system. Finally, the enhanced absorption of light in the material coupled to the antenna (in our case, a monolayer WSe$_2$), quantified by the excitation rate proportional to the intensity of the local near-field, $\gamma_{exc} \propto (|E|/|E_0|)^2$ (Fig. 1a, b), should in principle lead to a more efficient excitation of an SPE.

As shown in Fig. 1c–e, we carried out numerical simulations (see "Methods" and Supplementary Note I) to extract the values of these three parameters for a dipole emitting at $\lambda_{em} = 750$ nm coupled to either GaP dimer nanoantennas (data in red) or to SiO$_2$ nanopillars (dark blue). The dipole is placed at the edge of the dimers, where the interaction is maximized[7], and is aligned perpendicularly to the edge of the nanopillar (Supplementary Fig. 1). For the dimers with radii above 200 nm, the field enhancement at the outer edges is comparable to that in the gap (Supplementary Fig. 1a). As shown in Fig. 1c–e, a GaP nanoantenna may induce an enhancement of the PL intensity by at least two orders of magnitude[7,24,30], compared to SiO$_2$ nanopillars, as a consequence of the increase in both the spontaneous emission rate $F_p$ (Fig. 1c), the excitation rate $\gamma_{exc}$ (Fig. 1d), and a

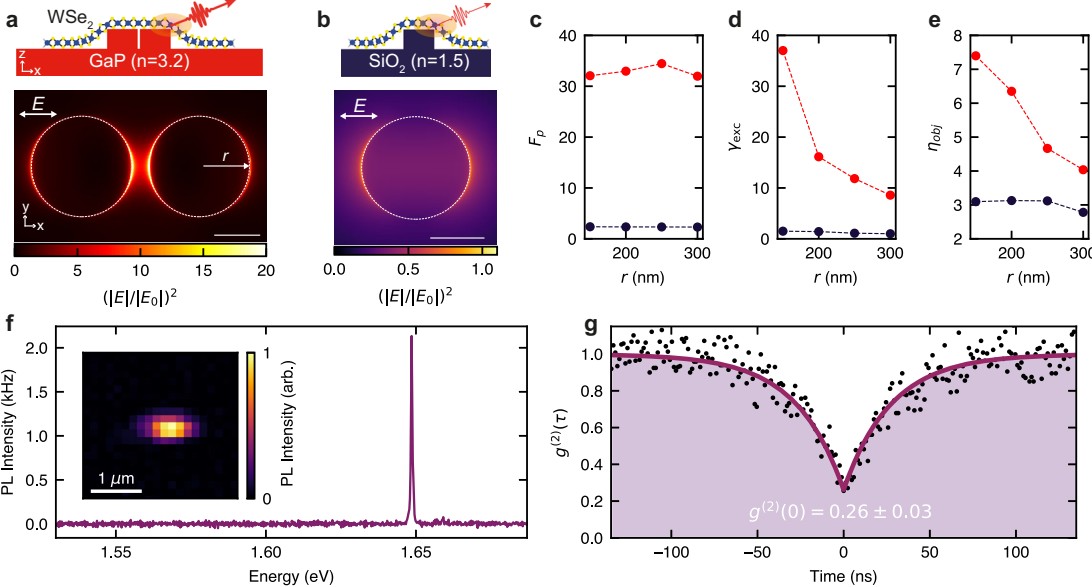

**Fig. 1 Optical properties of nanoantennas and single-photon emitters in monolayer WSe₂. a, b** Top panel: schematics of monolayer WSe₂ on top of GaP nanoantennas (**a**) and SiO₂ nanopillar (**b**). Lower panel: calculated relative intensity of the electric field in the scattered wave (E) over that in the incident wave ($E_O$). Results are shown for (**a**) a GaP dimer nano-antenna ($r = 150$ nm, $h = 200$ nm, gap $= 50$ nm) and (**b**) SiO₂ nanopillar ($r = 150$ nm, $h = 100$ nm). The calculated intensity is shown for the plane at the top of each structure. Scale bar: 150 nm. **c–e** The calculated Purcell enhancement factor (**c**), excitation rate (**d**), and light collection efficiency (**e**) for a dipole emitter placed at the position of the highest field enhancement shown in (**a**) and (**b**) of the GaP dimer nanoantennas (red) and SiO₂ nanopillars (dark blue) relative to the same parameters for this dipole in a vacuum. See further details in "Methods "and Supplementary Note I. **f** A $T = 4$ K PL spectrum of a WSe₂ SPE on a GaP nano-antenna ($r = 250$ nm, $h = 200$ nm, gap $= 50$ nm), excited` with a 638 nm pulsed laser with 20 MHz repetition rate and an average power of 15 nW. Inset: map of the integrated PL intensity of this monolayer ($T = 4$ K). **g** Second-order photon correlation curve for the PL signal in (**f**).

relatively modest effect in the collection efficiency $\eta_{\text{obj}}$ (Fig. 1e), as expected from the similar geometries of dimers and single pillars.

We note, that due to the high refractive index, for GaP most of the emitted light is directed downwards into the substrate, allowing only about 10% or less to be collected in the first lens (see further details in Supplementary Note 1). The highest single-photon rate of 69 MHz, which we quote in this work, corresponds to the total number of photons emitted by the SPE, whereas in the experiments reported below we measure only up to 5.5 MHz generation rate at the first lens.

In order to experimentally examine these effects, we transferred WSe₂ monolayers on top of an array of GaP nanoantennas (see "Methods" and Supplementary Note III) and on SiO₂ nanopillars as a reference (see Supplementary Note IV). With this approach, we achieve localized strain in the monolayer, introduced by the underlying nano-structure[8], which promotes the occurrence of localized SPEs at cryogenic temperatures. We find that the SPEs are formed on nearly all nanoantennas with a yield above 90%, in agreement with previous reports[19,20]. For some nanoantennas, we find multiple emitters (see Supplementary Note III).

The samples are placed in a gas exchange cryostat, at a temperature of $T = 4$ K, and excited (unless stated otherwise) with a non-resonant pulsed laser at 638 nm with 90 ps pulse width and a variable repetition rate. The laser excites below the GaP bandgap and is absorbed only by the WSe₂ monolayer. The inset in Fig. 1f shows a map of the integrated PL intensity from a WSe₂ monolayer deposited on top of a GaP dimer nano-antenna ($r = 250$ nm). The PL signal exhibits a strong localization at the nano-antenna position, with negligible emission from the surrounding area where the unstrained WSe₂ monolayer is positioned. As shown in Fig. 1f, we observe bright and narrow PL lines, with suppressed background PL from the band of localized states in WSe₂ as is usually observed when WSe₂ is deposited on

SiO₂ pillars (Supplementary Fig. 7a). We demonstrate the single-photon operation of the localized emitters in photon correlation measurements (see "Methods"). Figure 1g shows the second-order correlation function, $g^{(2)}(\tau)$, for the emitter shown in Fig. 1f, excited at $\lambda_{\text{exc}} = 725$ nm, ~35 meV below the WSe₂ A-exciton resonance. The pronounced anti-bunching behavior at zero time delay exhibits $g^{(2)}(0) = 0.26 \pm 0.03$, confirming the nonclassical photon emission statistics. In Supplementary Note V, we further correlate the SPEs emission energy to the strain induced in the 2D layer by the nanoantennas. Stretching of the WSe₂ monolayer results in a progressively larger red-shift of the SPE emission when deposited on nanoantennas with smaller radii[8]. This behavior, analogous to the red-shift of WSe₂ excitons under tensile deformation[8], confirms the impact of strain on the confinement potential and emission energy of WSe₂ SPEs.

The position of the SPE and the orientation of the emitter dipole relative to the nano-antenna are important factors for the PL enhancement[7]. We expect that the SPEs will tend to form naturally around the edges of the nanopillars (Fig. 1a), where both the tensile strain and photonic enhancement are maximized as follows from our previously reported theoretical and experimental results (see ref. [8] and Supplementary Note V). In these outer edge positions, the SPEs still experience a strong enhancement of the electric field, which for the nanopillars with the radii above 200 nm is comparable with the enhancement in the gap between the pillars (see Supplementary Fig. 1a).

In our experiments (not reported here), we find that WSe₂ SPEs on planar SiO₂ have comparable PL intensities and lifetimes to those formed on SiO₂ pillars, with the latter having the advantage of controlled positioning. Furthermore, we did not observe SPEs of reliably measurable PL intensity on planar GaP, where we find that overall PL of WSe₂ is quenched as seen for example in Fig. 1f. In what follows, we focus on GaP dimer

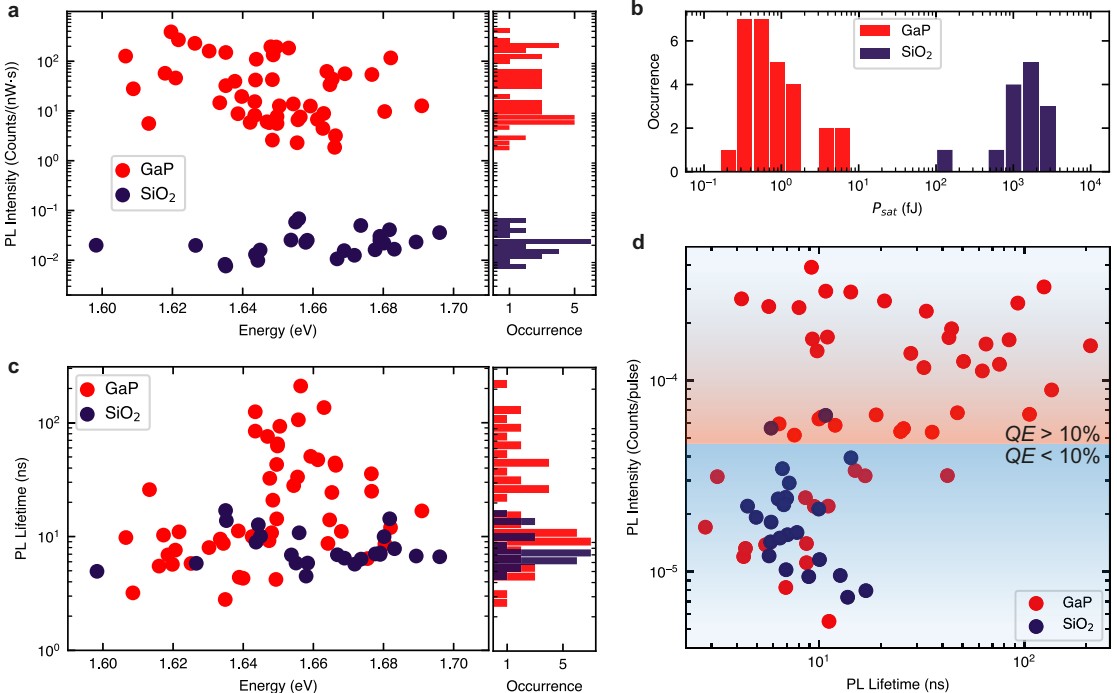

**Fig. 2 Comparison of optical properties of SPEs formed on GaP dimer nanoantennas and on SiO$_2$ nanopillars.** Data for SPEs on GaP dimer nanoantennas are shown in red and for SPEs on SiO$_2$ nanopillars is shown in dark blue. **a** PL intensity normalized by the average laser excitation power measured in the SPE saturation regime. The histogram on the right shows the occurrence of the observed PL intensity values. **b** Energy per laser pulse required for SPE saturation, $P_{sat}$. See Supplementary Fig. 4 for more details on how $P_{sat}$ is extracted from the PL data. **c** PL decay times (main plot) and its occurrence (right). **d** SPE PL peak intensity divided by the laser repetition rate plotted versus SPE PL decay time. The red and blue areas of the plot correspond to SPEs with $QE > 10\%$ and $QE < 10\%$, respectively. See main text and Supplementary Note VI for more details of how $QE$ was estimated.

nanoantennas, as these provide pronounced and interesting photonic effects, as was shown in our preliminary work[7,8]. On the other hand, less studied GaP monomers (single nanopillars) may also be useful for achieving SPE positioning and improved PL in WSe$_2$ monolayers. This may be a subject of another investigation beyond our current work.

**Quantum efficiency enhancement of SPEs on GaP nanoantennas.** We analyzed more than 50 SPEs on GaP dimer nanoantennas, with radii ranging from 150 nm up to 300 nm, selecting localized WSe$_2$ emitters with sub-meV linewidths. For these emitters, we observed common features such as linearly polarized emission, saturation of the PL intensity under increased excitation power, as well as PL lifetimes in the ns range (see Supplementary Note III). We observed no preferential orientation in the SPEs polarization, with different orientations of polarization even for emitters created on the same nano-antenna. Figure 2a shows the values of the PL intensity for SPEs on GaP nanoantennas (red dots) and on SiO$_2$ nanopillars (dark blue dots), acquired in the PL saturation regime and normalized to the average excitation pump power. The plot also shows the PL peak position for each studied SPE, where no correlation between the intensity and spectral position is observed. For SPEs coupled to GaP nanoantennas, we observe from two to four orders of magnitude higher power-normalized PL intensity, compared to SPEs found on SiO$_2$ nanopillars. Further insight into this behavior is provided by the SPE PL saturation powers, presented in Fig. 2b. Since we used different repetition rates from 5 to 80 MHz in these measurements due to a large variation in the PL lifetimes, in Fig. 2b we plot the energy per pulse value $P_{sat}$, defined as the time-integrated average power divided by the laser repetition rate. We readily observe more than three orders of magnitude lower saturation pulse energies for the emitters on GaP nanoantennas.

In our case, 1 fJ pulse energy corresponds to the energy density per pulse of 30 nJ/cm$^2$. Nonetheless, for such low powers, the SPEs coupled to GaP nanoantennas provide some of the highest counts per second (> 30,000) so far observed in TMD monolayers. In what follows, we will consider the factors that could contribute to this observation.

One of the obvious factors, expected to contribute to the reduced values of $P_{sat}$ is the enhanced absorption rate ($\gamma_{exc}$) in WSe$_2$ monolayers coupled to GaP nanoantennas. However, this can only account for a reduction of the saturation power of up to 40 times as predicted by our simulations (Fig. 1d). A similar maximum enhancement for the power-normalized PL intensity may be expected due to the enhanced $\gamma_{exc}$. Thus, additional factors need to be considered, mostly linked to the exciton dynamics and $QE$ of the combined 2D-WSe$_2$/SPE system.

In Fig. 2c, we compare PL lifetimes of SPEs on GaP and on SiO$_2$. For the SPEs on the SiO$_2$ nanopillars, we observe lifetimes of the order of 10 ns, consistent with previous reports[19,20]. On the contrary, the SPEs on the GaP nanoantennas exhibit a broad distribution of lifetime values, ranging from 2 ns up to more than 200 ns. The radiative and nonradiative population decay rates in WSe$_2$ SPEs ($\Gamma_{nr}$ and $\Gamma_r$ in Fig. 3d) are dependent on the shape and confinement energy of the strain potential, and the PL decay dynamics is defined by the relationship between them: if one of the rates is much higher than the other, it will define the PL decay time.

In order to shed light on the relationship between these rates, in Fig. 2d we plot the SPEs fluorescence lifetime intensity distribution[32]. The SPEs on SiO$_2$ exhibit low PL emission with relatively short lifetimes (blue area). In SPEs coupled to GaP nanoantennas, we observe either a much higher PL intensity and similar lifetimes, or longer lifetimes with comparable brightness (red area in Fig. 2d). The $QE$ of an SPE under pulsed excitation

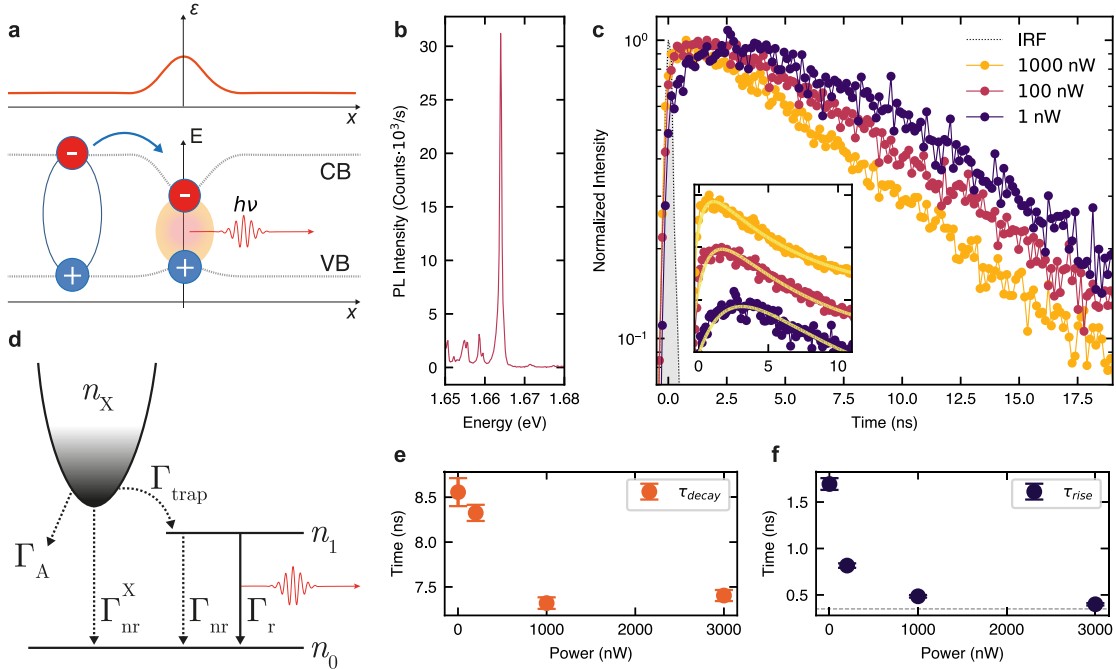

**Fig. 3 PL dynamics in SPEs coupled to GaP nanoantennas. a** Schematic showing the conduction (CB) and valence band (VB) behavior as a function of strain (shown in the top panel with a red line), and a single exciton trapping from the reservoir in 2D WSe₂ into a strain-induced potential minimum, giving rise to the nonclassical light red-shifted from the emission in the unstrained monolayer. **b** Time-integrated PL spectrum of the strain-induced WSe₂ SPE with a QE of 86% and saturation power of 57 nW for 80 MHz repetition rate, for which the data in (**c**), (**e**), and (**f**) are presented. **c** PL decay curves for the SPE in (**b**) measured at different laser powers. The instrument response function (IRF) is shown in gray. Inset: zoom-in of the PL traces also showing fitting with the analytical model discussed in Supplementary Note VII. The PL traces in the inset are plotted on a linear scale and shifted vertically for display purposes. **d** Schematic of the three-level system representing the dark exciton reservoir ($n_X$), the exciton in the SPE ($n_1$) and the ground level ($n_0$), and the processes describing radiative and nonradiative decay of $n_X$ and $n_1$ populations. See text for more details. **e**, **f** PL decay (**e**) and PL rise (**f**) times, as a function of the excitation power, obtained from the data fitting in Fig. 3c. For 3000 nW, $\tau_{\mathrm{rise}}$ approaches the instrument resolution (gray dashed line). The error bars are calculated from the standard deviation of the fit values.

can be estimated from the number of detected photons at saturation, divided by the laser repetition rate[21]. After taking into account the losses of the experimental setup and the collection efficiency of the nano-antenna from numerical simulations (see additional details in Supplementary Note VI), we estimate an average QE for SPEs coupled to GaP nanoantennas of 21 ± 3%, with a maximum value reaching 86%. For SPEs on SiO₂ nanopillars, we estimate an average QE of 4 ± 2% consistent with the previous reports[21].

We thus conclude that the PL decay times of SPEs on SiO₂ nanopillars are mainly defined by nonradiative processes (i.e., $\Gamma_{\mathrm{nr}} \gg \Gamma_r$), and that the true radiative lifetimes should by far exceed the measured decay times of ≈ 10 ns. On the other hand, for the SPEs on GaP nanoantennas exhibiting high QE, the lifetimes are mostly defined by the radiative decay (i.e., $\Gamma_{\mathrm{nr}} < \Gamma_r$ or $\Gamma_{\mathrm{nr}} \ll \Gamma_r$), which, as we can conclude from Fig. 2d, vary between 2 and 200 ns. Comparing this with the SPEs on SiO₂, we can conclude that the high QE SPEs on GaP exhibiting lifetimes of the order of 10 ns or shorter are most likely affected by the Purcell enhancement increasing their radiative rates, and thus are possibly positioned in the near-field hotspots.

The high QE SPEs with PL decay times > 10 ns clearly must experience much slower nonradiative processes than SPEs on SiO₂, as the nonradiative lifetimes must be slower than the measured PL decay times. This is also in contrast to previously reported SPEs coupled to plasmonic structures[21], where despite the very large Purcell enhancement and sub-ns PL lifetimes, the maximum QE of 12.6% was reported for WSe₂ monolayers extracted similarly to our work from bulk crystals grown by

chemical vapor transport. This implies high nonradiative rates in this system ($\Gamma_{\mathrm{nr}} > \Gamma_r$). On the other hand, SPEs with PL lifetimes in the range of 100 ns were previously observed only in monolayer WSe₂ encapsulated in hexagonal boron nitride[12], known for suppressing the nonradiative processes. In our case, possible causes for suppression of nonradiative decay could be the high surface quality of crystalline GaP structures, or that some SPEs are formed in the suspended parts of the monolayer in proximity to the near-field hotspots and between the pillars[8]. We cannot exclude that some of the high QE SPEs with PL decay times > 10 ns still experience Purcell enhancement, implying that the true radiative times in some WSe₂ SPEs may reach hundreds of ns. Further detailed insight in the PL dynamics is given below in the discussion of Fig. 3.

**Dynamics of exciton formation in strain-induced SPEs.** SPEs in WSe₂ are attributed to the occurrence of strain-induced local potential minima[20,33], essentially zero-dimensional (0D), that can host a small number of confined excitons, similar for example to semiconductor quantum dots[34]. Contrary to other group-VI TMDs, tensile strain in WSe₂ results in the lowering of the conduction band (CB) minimum and the rise of the valence band (VB) maximum, as shown in Fig. 3a, both located at the K points in the momentum space[8,35]. This creates an energy landscape where a very small fraction of the 2D exciton population may be captured into such 0D centers, giving rise to nonclassical light emission from confined states, at photon energies lower than that of both bright and dark excitons in unstrained WSe₂.

As shown in Fig. 2, in the case of WSe₂ placed on GaP nanoantennas, both the quantum yield and brightness of the SPEs are greatly enhanced, allowing new insight into the exciton dynamics in this hybrid 2D–0D system. Figure 3b shows a PL spectrum for an SPE exhibiting $QE$ of 86 ± 3%. Figure 3c shows the time-resolved PL decay for the same SPE measured with 20 MHz repetition rate. The PL decay curves are obtained at different powers of 1, 100, and 1000 nW considerably below, close, and considerably above the saturation power, respectively. For clarity, the inset zooms in on the short times after the laser pulse excitation. At low power, we clearly observe a ns-scale rise time, which shortens as the power is increased also accompanied by a relatively weak shortening of the PL decay time.

We fit the data with a simple empirical model assuming an exciton reservoir, which feeds excitons into the SPE. The model can be solved analytically (see Supplementary Note VII for more details) and is used to fit the data, as shown in the inset of Fig. 3c, providing rise and decay times plotted in Fig. 3e and f. Here, we see that as the power is increased, the rise time, $\tau_{\mathrm{rise}}$, changes strongly from 1.7 ns to times approaching the experimental resolution, whereas the PL decay time $\tau_{\mathrm{decay}}$ decreases from 8.5 to 7.4 ns.

In order to understand this behavior, we consider several processes, which influence both the populations of the high energy 2D exciton reservoir and the SPE itself. We argue that the exciton reservoir with the population $n_{\mathrm{X}}$ in Fig. 3d corresponds to the population of dark excitons, which we infer from the very slow PL rise time of 1.7 ns at low power, in contrast to the expected lifetime of the bright excitons of a few ps[36,37]. The dark excitons decay mostly via sample-specific nonradiative recombination with a rate $\Gamma_{\mathrm{nr}}^{\mathrm{X}}$ and, importantly, via the exciton–exciton (Auger) annihilation[28,29], which grows with the increasing power as $\Gamma_{A} n_{\mathrm{X}}^{2}$. Trapping of dark excitons with a rate $\Gamma_{\mathrm{trap}}$ into the strain-induced SPE is responsible for a negligible reduction of $n_{X}$, as the anti-bunching photon emission implies that only one exciton per laser excitation cycle can be created in the SPE. We thus also introduce a probability $n_{1}$ for the SPE to be filled with an exciton with $0 \leq n_{1} \leq 1$. The trapping of the dark excitons is the only source of the SPE population, and is included as a term $\Gamma_{\mathrm{trap}}(1 - n_{1})n_{\mathrm{X}}$ in the equations below. Here, we take into account the effect of the SPE occupancy on the reduced efficiency of the exciton trapping with the factor $(1 - n_{1})$, providing one of the mechanisms for the PL saturation with increasing power observed in the experiment. The population of the SPE decays radiatively and non-radiatively with rates $\Gamma_{\mathrm{r}}$ and $\Gamma_{\mathrm{nr}}$, respectively. Here, for simplicity, we neglect the SPE's internal confined-state structure, which we uncover in PL excitation experiments (see Fig. 4 for details). The rate equations capturing the behavior of the three-level system depicted in Fig. 3d are shown below:

$$\frac{dn_{\mathrm{X}}}{dt} = -[\Gamma_{\mathrm{nr}}^{\mathrm{X}} + \Gamma_{\mathrm{trap}}(1 - n_{1})]n_{\mathrm{X}} - \Gamma_{A} n_{\mathrm{X}}^{2} \quad (1)$$

$$\frac{dn_{1}}{dt} = -(\Gamma_{\mathrm{r}} + \Gamma_{\mathrm{nr}})n_{1} + \Gamma_{\mathrm{trap}}(1 - n_{1})n_{\mathrm{X}} \quad (2)$$

We estimate that for 1 nW laser power at 20 MHz repetition rate and 5% light absorption in WSe₂, the dark exciton density $n_{x} \approx 3 \times 10^{8}$ cm⁻² will be created. This is probably the lower bound, as the near-field electric field enhancement can locally lead to the increase of this value by a factor exceeding 10. At this low-power limit, the Auger annihilation can be neglected[28] and the unsaturated SPE emission leads to an average (per pulse) $n_{1} \ll 1$. The PL rise dynamics is then defined by the predominately nonradiative decay of the dark exciton reservoir with the rate $\Gamma_{\mathrm{nr}}^{\mathrm{X}}$. As the power is increased, and both $n_{\mathrm{X}}$ and $n_{1}$ grow, two

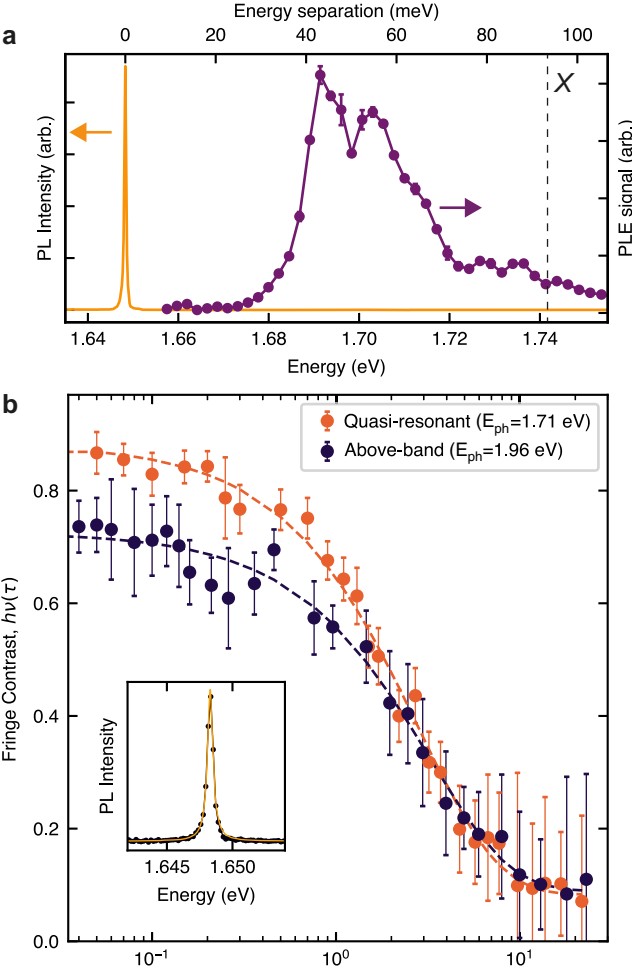

**Fig. 4 Measurements of SPE coherence. a** PL emission (yellow) and the PL excitation (PLE) spectra (purple) from a WSe₂ SPE. The PLE signal exhibits a series of confined exciton peaks below the neutral exciton PL peak position (X). **b** Interference fringe contrast, $\nu(\tau)$ for two different excitation laser wavelengths. The dependence for the above-band excitation (dark blue dots) is obtained at 1.96 eV (638 nm) and yields a coherence time of $T_2 = 2.83 \pm 0.20$ (fitting shown with a dashed line). The dependence for the quasi-resonant excitation (orange dots) is obtained with a laser tuned to 1.71 eV (725 nm), lower than the A-exciton in unstrained monolayer WSe₂, yielding a coherence time of $T_2 = 3.12 \pm 0.40$. The error bars are calculated form the standard deviation of the fit values. Inset: SPE PL spectrum under the quasi-resonant excitation (black dots) fitted with a single Lorentzian peak (yellow solid line) with a linewidth of 450 µeV, corresponding to a coherence time of $T_2 = 2.9$ ps, in close agreement with the values obtained from the interferometric measurements.

additional processes become important: the Auger annihilation described by the term $\Gamma_{A} n_{\mathrm{X}}^{2}$ and the saturation of the SPE with the corresponding term $\Gamma_{\mathrm{trap}}(1 - n_{1})n_{\mathrm{X}}$. For the powers presented in Fig. 3, $n_{\mathrm{X}}(t = 0)$ is estimated to be of the order of $10^{10}$ cm⁻² for the power of 100 nW and $10^{11}$ cm⁻² for 1000 nW, in the range where the Auger annihilation was found to be very efficient[28,29].

While a more detailed study at low powers could help to separate the contributions from the Auger annihilation and SPE saturation, it is possible that in the high power regime the SPE PL saturation is influenced not only by the state-filling effect but also by the nonradiative depletion of the dark exciton reservoir. In the case of the bright and high $QE$ SPEs in WSe₂/GaP nano-antenna system, high photon counts can be achieved at low-excitation

powers, thus circumventing the requirement for increased pumping. On the other hand, in the SPEs in WSe$_2$ on SiO$_2$ nanopillars, where both the QE and brightness are low, increased pumping is required to observe the SPE PL. This has a negative effect on the population of the reservoir via the Auger annihilation and thus, through such negative feedback, leads to the requirement to further increase the power. Eventually, both the low QE and its further reduction due to the Auger annihilation lead to a very large three order of magnitude increase in the saturation powers in the SPEs in WSe$_2$ on SiO$_2$ nanopillars compared with those on GaP nanoantennas, as seen in Fig. 2. In support of these conclusions, we also note that high saturation powers, similar to those observed by us in WSe$_2$ SPEs on SiO$_2$ were also reported in SPEs coupled to plasmonic structures[21], where very fast nonradiative processes in the 2D WSe$_2$ should be expected.

**Coherence of a strain-induced SPE.** The coherence of WSe$_2$ SPEs has been previously investigated only under high power densities and non-resonant excitation[21]. Here, we evaluated the first-order correlation function, $g^{(1)}(\tau)$, for the SPE shown in Fig. 4a, in a Mach–Zender interferometer setup[38] and compared different excitation schemes (see "Methods"). We employed an above-band excitation using a 1.96 eV (638 nm) cw laser, corresponding to energy higher than the A-exciton resonance in monolayer WSe$_2$ (dashed line in Fig. 4a). Under these conditions, high energy excitons are created in the continuum of states above the excitonic resonance, introducing dephasing for instance via scattering with phonons and impurities or via exciton–exciton interactions. To reduce the impact of such processes, we also used a quasi-resonant excitation with a cw laser at 1.71 eV (725 nm). As shown in Fig. 4a, this excitation is resonant with higher energy states within the SPE[1]. Figure 4b shows the measured fringe contrast, $v(\tau)$, of the WSe$_2$ SPE under the two excitation schemes (see "Methods"). By fitting the observed decay of the fringe contrast with a single exponential decay function, $g^{(1)}(\tau) \approx \exp(-|\tau|/T_2)$, we extract a coherence time of $T_2 = 3.12 \pm 0.40$ ps under quasi-resonant excitation, and of $T_2 = 2.83 \pm 0.20$ ps for above-band excitation. The differences between the excitation schemes have a negligible effect on the SPE dephasing time, implying a complex relaxation process within the confined states of the SPE. We find that the PL full-width at half maximum (FWHM) of $\approx 450\,\mu eV$ corresponds to $T_2 = 2.9$ ps (FWHM $= 2\Gamma = 2\hbar/T_2$) close to the observed $T_2$ values, indicating that the coherence of the studied SPE is limited by pure dephasing, which we attribute to interactions with phonons during the exciton relaxation[39], as for the excitation power < 20 nW used in the experiment the contribution of the Auger annihilation can be excluded. The observed SPE $T_2$ values are one order of magnitude higher than those reported for monolayer WSe$_2$ of 0.3 ps[39]. Excitation in resonance with the lowest energy optical transition in the SPE could be employed to gain access to the intrinsic coherence times of the confined excitons.

## Discussion

In summary, we have demonstrated that high-refractive-index GaP nanoantennas offer an efficient approach for nano-scale positioning and QE enhancement in strain-induced SPEs in monolayer WSe$_2$. We found $10^2$ to $10^4$ enhancement of the PL intensity for WSe$_2$ SPEs coupled to GaP nanoantennas compared with those formed on low-refractive-index SiO$_2$ nanopillars. We demonstrate that this is primarily caused by the greatly increased QE in the 2D/0D WSe$_2$ system coupled to GaP nanoantennas arising from the enhancement of the radiative rates in such SPEs through the Purcell effect, as well as the reduction of the

nonradiative decay rates. Importantly, this allows bright emission from the SPEs to be excited with energy densities per laser pulse below 30 nJ/cm$^2$ corresponding to the energy per pulse below 1 fJ, enabling the SPE operation at low exciton densities in the 2D WSe$_2$, thus avoiding the exciton–exciton annihilation. The powers at which SPEs on GaP nanoantennas provide bright emission are approximately three orders of magnitude below those required for the operation of the SPEs on SiO$_2$ pillars studied in this work, as well as those previously reported for SPEs formed on plasmonic nanostructures[21], despite the large Purcell enhancement factors found in the latter system[21]. Further improvement and consistency of the operation of SPEs can possibly be achieved by employing deterministic defect placement[40,41], while the required excitation powers can be further reduced by employing much cleaner WSe$_2$ grown by the so-called flux technique[21]. The photon collection efficiency in our approach can be potentially improved by engineering the nanocavity geometry and materials, for instance by using nanoantennas made from high-index TMDs[42,43]. These materials can be deposited on any type of substrate. By placing them on a metallic mirror similarly to the strategy pursued by Luo et al. in ref. [21], most of the light can be redirected upward, and the collection efficiency in the first lens can be considerably increased. Overall, our work suggests that hybrid systems composed of 2D semiconductors coupled to dielectric nanoantennas are a powerful means for controlling quantum light generation.

## Methods

**Sample fabrication.** The GaP dimer nanoantennas were fabricated using electron-beam lithography, followed by several wet and dry etching steps as described in ref. [24]. Arrays of nanoantennas separated by 4 μm were made. The dimers had a gap of ≈ 50 nm, a height of 200 nm, and nanopillar radii (r) of 150, 200, 250, and 300 nm. Atomically thin monolayers of WSe$_2$ were mechanically exfoliated from commercially available bulk single crystals (HQ Graphene) onto poly-dimethylsiloxane (PDMS) polymer substrates. The monolayer thickness was identified by examining room temperature PL with an imaging method described in ref. [44]. The monolayers were then transferred on top of the GaP nano-antenna array, by using the same PDMS substrates, with an all-dry transfer technique in a home-built transfer setup[45].

**Optical spectroscopy.** Low-temperature PL spectroscopy was carried out with a sample placed in low-pressure He exchange gas within a confocal microscope platform allowing free space optical access and high precision sample positioning (Attocube). The whole microscope stick was inserted in a liquid helium transport dewar (Cryo Anlagenbau Gmbh) and a nominal sample temperature of 4 K was used in all reported experiments. The excitation from the lasers used in this work was delivered through single-mode fibers to the optical breadboard placed at the top of the microscope stick, where it was collimated and directed onto the sample through a window at the top of the stick. For pulsed excitation, we used a diode laser (PicoQuant) at 638 nm, with a variable repetition rate from 5 to 80 MHz and a pulse width of 90 ps. For continuous-wave excitation, we used a tunable Ti-Sapphire laser (M Squared SOLSTIS). PL emitted by the sample was collected with an aspheric lens (NA = 0.64) and coupled at the breadboard into a single-mode optical fiber, which delivered it to a spectrometer (Princeton Instruments SP2750), where it was detected with high-sensitivity liquid nitrogen cooled charge-coupled device (Princeton Instruments PyLoN). For the time-resolved spectroscopy, the PL was also sent through the spectrometer to another exit port, where it was measured with an avalanche photodiode (ID100-MMF50) connected to a photon-counting card (Becker and Hickl SP-130). A Hanbury Brown-Twiss setup used for the evaluation of the second-order correlation function ($g^{(2)}(\tau)$) was equipped with two superconducting nanowire single-photon detectors (Single Quantum) and a similar photon-counting card. The SPE was excited with a continuous-wave diode laser (Thorlabs HL7302MG) at 730 nm with a power of 2 nW. Light from the SPE was directed to the nanowire detectors using a multimode fiber. The emitted light from the SPE was filtered with a pair of filters (Thorlabs FESH800 and Thorlabs FELH750) providing a transparency window between 750 and 800 nm.

**Coherence measurements.** For evaluation of the first-order correlation function ($g^{(1)}(\tau)$), we used a Mach–Zender interferometer setup[38] with a phase shifter in one arm and a variable optical delay in the other. By sweeping the voltage of the phase shifter, the interference of light emitted by the SPE was measured using an avalanche photodiode at one output port of the interferometer. By measuring the intensity at the local maxima ($I_{max}$) and minima ($I_{min}$) of the interference fringes,

we evaluate the fringe contrast ($\nu$) as:

$$\nu = \frac{I_{\max} - I_{\min}}{I_{\max} + I_{\min}} \quad (3)$$

This procedure was repeated for increasing delay times until the fringes were no longer resolved. The relationship between the fringe contrast and the first-order correlation function is given by the following equation:

$$\nu(\tau) = (1 - \epsilon) \frac{|g^{(1)}(\tau)|}{g^{(1)}(0)} \quad (4)$$

where $1 - \epsilon$ is the maximum resolvable fringe contrast in the setup and $|g^{(1)}(\tau)|$ is the first-order correlation function, excluding the fast oscillations at the emitter frequency.[38] In the absence of any spectral diffusion, the fringe contrast as a function of time follows a single exponential decay, with an exponential fit allowing the evaluation of the coherence time ($T_2$) of the emitter.

**Simulations**. The distributions of the electric field in Fig. 1 were calculated with a commercial finite-difference time-domain software (Lumerical Inc.). In the simulations, we illuminated the structure with a linearly polarized plane wave at $\lambda = 750$ nm with a normal incidence from the vacuum side of the substrate. See Supplementary Note I for further details on the simulations.

## Data availability
The data that support the findings of this study are available from the corresponding author upon request.

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

## Acknowledgements
L.S., P.G.Z., and A.I.T. thank the financial support of the European Graphene Flagship Project under grant agreements 881603 and EPSRC grant EP/S030751/1. L.S. and A.I.T. thank the European Union's Horizon 2020 research and innovation programme under ITN Spin-NANO Marie Sklodowska-Curie grant agreement no. 676108. P.G.Z. and A.I.T. thank the European Union's Horizon 2020 research and innovation programme under ITN 4PHOTON Marie Sklodowska-Curie grant agreement no. 721394. J.C., S.A.M., and R.S. acknowledge funding by EPSRC (EP/P033369 and EP/M013812). C.L.P., A.J.B., A.I.T., and A.M.F. acknowledge funding by EPSRC Programme Grant EP/N031776/1. S.A.M. acknowledges the Lee-Lucas Chair in Physics, the Solar Energies go Hybrid (SolTech) programme, and the Deutsche Forschungsgemeinschaft (DFG, German Research Foundation) under Germany's Excellence Strategy - EXC 2089/1 - 390776260.

## Author contributions
L.S., P.G.Z., A.I.T., S.A.M., and R.S. conceived the idea of the experiment. L.S. and P.G.Z. fabricated WSe$_2$ layers, transferred them on GaP nanoantennas, and carried out

numerical modeling. L.S., P.G.Z., C.L.P., and A.J.B. carried out optical spectroscopy measurements on WSe$_2$. J.C. fabricated GaP nanoantennas. J.C. and R.S. designed GaP nanoantennas. L.S. and E.M. designed and analyzed the rate equation model. L.S., P.G.Z., C.L.P., A.J.B., and A.I.T. analyzed optical spectroscopy data. S.A.M., R.S., A.M.F., and A.I.T. managed various aspects of the project. L.S., P.G.Z., and A.I.T. wrote the manuscript with contributions from all co-authors. A.I.T. oversaw the whole project.

## Competing interests

The authors declare no competing interests.
