## [Peer Review File · Nature Communications]

Reviewers' Comments:

Reviewer #1:

Remarks to the Author:

Luca Sortino et al. reported WSe₂ single-photon emitters coupled to the GaP nano-antennas. The structure, i.e. WSe₂ on top of GaP dimer, is the same geometry as in their previous paper published in the Nature communication, 2019. However, this manuscript is different from the previous paper in several aspects: 1) single-photon generation, 2) Quantum efficiency enhancement. The paper also includes insightful explanation to back up their experimental observations. As a result, the paper provides much new information on SPEs from WSe₂.

Therefore, I recommend this paper to be published in Nature communication after minor revision.

1. Authors claimed the brightest single-photon emitter among TMD SPEs. It will be convincing if the authors include a table to compare the brightness from previous papers.

2. SPEs due to strain

2-1. What was the yield to find single-photon emitters on GaP or SiO structures? It will be clear if a PL map is provided to visualize this, i.e. array of GaP (or SiO₂) structures with SPEs created in an array.

2-2. For GaP dimer structure, where is the maximum tensile strain? Is it along the edge of the dimer structure? Is there a strain analysis on this structure? How likely is it that SPEs created at the hotspot?

3. Polarization of SPEs

Was there a tendency in polarization from strain-induced SPEs? If so, are they preferably x-pol or y-pol?

4. Longer lifetime from SPEs in GaP

An explanation for a longer lifetime with GaP dimer was briefly touched in the manuscript, however, it seems not easy to follow the logic by reading the manuscript. It would be great if authors can explain Figure 2(d) using the diagram shown in Figure 3(d)

The lifetime of SPEs is dominated by faster decay between Γ_{nr} and Γ_r (as the authors also mentioned). So if Γ_r is accelerated, i.e. radiative decay is faster than non-radiative one, the radiated emission now become a major decay channel. However, in this case, the lifetime of an emitter should be shorter than the non-radiated decay lifetime. Therefore, with Γ_{nr} present, how can a lifetime get longer due to the Purcell effect?

5. Minor correction: blue data points throughout figures are not very visible (it looks either black or purple)

Reviewer #2:

Remarks to the Author:

The manuscript by Sortino et al. reports on the coupling of single-photon emitters in WSe₂ to dielectric nanoantennas. The coupling increases the quantum efficiency from 4% to up to 86% and the PL brightness by a factor of 100 to 10000. What is new compared to other works, is the coupling to a *dielectric* resonator (compared to plasmonic structures in most other works), and thereby achieving a very high brightness of the single photon emission. The manuscript give a detailed insight into the physical origin and processes of the single photon emission in atomically thin WSe₂.

The manuscript is well written, although I see some space for improvements (see comments), however, all points are only minor and should be easily addressable by the authors. It is technically sound and all conclusions and claims are supported by the data.

The most difficult question is now, how significant are the results. Some of the results, e.g. the increased PL are nice, but I would not call it a major breakthrough. Others are completely new for this kind of single-photon emitter, e.g. the coherence measurements and the detailed analysis of the dynamics.

In conclusions, I think that this manuscript is an important step in understanding the SPE in 2D semiconductors. So, my feeling is, that it is important for the community, but maybe not of too much importance for the wider field.

Comments/Minor issues:

1. Enhancement by the dimer antennas: In Fig. 1 a-e the properties of the GaP and SiO₂ structures are shown, and obviously, the dimer is far superior. However, the figure gives not all information directly.

a) Where is the maximum of the E-field enhancement exactly (at the outer edge or the edges forming the gap)? From the color plot, I believe the latter, while schematic drawing suggest the outer edge. This should be marked in the plots.

[Later on, I found in the SI, that this could depend on the radius of the pillar, so the position changes in plot c-e?]

b) Since it is unpredictable where the SPE are formed (or activated) and which polarization they will have, I would be good to show also the enhancement for different special positions and the perpendicular polarization on the nanoantennas.

c) The antibunching dip is below 0.5, but not overwhelming. I don't see the reason why (the spectrum looks very clean). Can the authors discuss this. Do they find emitters with a better antibunching?

2. When looking on the many complex fabrication steps of creating plasmonic or dielectric nanostructures, the additional dielectric spacer for plasmonic nanoantennas seems not a convincing argument of increased fabrication complexity (p2, l25). The reduced coupling and not lossless nature of the plasmonic part are convincing enough.

3. Being a little bit pedantic, but the HBT setup was deceived by Robert Hanbury Brown and Richard Twiss, so the setup is called "Hanbury Brown and Twiss" or "Hanbury Brown-Twiss" setup. (At least without the dash between "Hanbury" and "Brown")

4. When I first read the SI Note VI, I was confused what the optical losses by the "optical breadboard" are (my first idea: That's an aluminum plate with threads). Eventually I think the authors refer to the optical components on the optical breadboard, and that the efficiency is only 50% let me think there is a beam-splitter involved. But I would appreciate if the authors can clarify this, and add a schematic drawing of the whole setup. It would also be good to estimate the error here.

5. No issue but just a question out of curiosity: Why do the authors use a single mode fibre here, where they lose 98% of their photons. A 50um multi-mode fibre could easily collect 90% of all photons and would only slightly decrease the spatial resolution (which seems uncritical here).

Reviewer #3:

Remarks to the Author:

The manuscript "Bright single photon emitters with enhanced quantum efficiency in a two dimensional semiconductor coupled with dielectric nano antennas" by L. Sortino et al., presents an interesting study on the incorporation of photonic enhancement through dielectric dimers that are used simultaneously to induce strain in WSe₂ flakes to create SPEs. The authors show a significant enhancement in photoluminescence brightness compared to pillars made of SiO₂. I think the findings of the manuscript are of interest to the generation and engineering of better quality single-photon emitters in 2D transition metal dichalcogenides. However, there is some ambiguity regarding the origins of such enhancement that I believe needs addressing.

Due to the lack of any experimental quantification of the dimer's optical response, it doesn't seem easy to fully understand the origins of the enhancement reported by the authors. For example, how does the performance of the SPEs on GaP dimers compare to the random SPEs in a flake on a bare GaP substrate or that on a GaP pillar without increased field enhancement caused by the dimer? That would be very important to know and fully quantify since the brightness of TMD emitters has been reported to vary based on the substrate materials as well as encapsulation, through hBN, for example. Having this information would make claims about enhancements in

brightness being due to Purcell enhancement more convincing.

There is no much discussion in the paper about the quantification of the stress induced by the dimer and whether the structural properties of the dimer have been selected for optimized stress or for their optical response. Is the maximum stress expected to happen, as in the pillars, on the edges, or in the gap region where the field enhancement is maximum? This would play a role in the various calculations in the manuscript since the points of maximum optical enhancements don't necessarily coincide with the SPEs.

I suggest including a better or zoomed up SEM image of the fabricated dimer as the one presented in Fig. S3 seems to show a low-quality structure. It would be insightful to characterize the scattering cross-sections of the actual dimers to confirm their performance matches that predicted from simulations.

What is the reason the localized SPEs spots on SiO₂ pillars can't be resolved in Fig. S7? Figure 1f suggests that the authors should be able to resolve the emitters. Also, what is the yield for SPE creation per GaP dimer? A similar PL map on dimers would be helpful to include.

Dear Editor,

We would like to resubmit our manuscript by Sortino L. et al. "Bright single-photon emitters with enhanced quantum efficiency in a two-dimensional semiconductor coupled with dielectric nano-antennas" in Nature Communications.

We trust that we have addressed all the reviewers' comments, as we discuss in detail below with a step-by-step response. We carefully revised the manuscript according to their concerns, as described in the responses below. All modified text in the manuscript is now given in red.

REVIEWER COMMENTS

Reviewer #1 (Remarks to the Author):

Luca Sortino et al. reported WSe₂ single-photon emitters coupled to the GaP nano-antennas. The structure, i.e. WSe₂ on top of GaP dimer, is the same geometry as in their previous paper published in the Nature communication, 2019. However, this manuscript is different from the previous paper in several aspects: 1) single-photon generation, 2) Quantum efficiency enhancement. The paper also includes insightful explanation to back up their experimental observations. As a result, the paper provides much new information on SPEs from WSe₂. Therefore, I recommend this paper to be published in Nature communication after minor revision.

Response: We are very grateful to the reviewer for acknowledging the value of our work and the recommendation for publication. We provide a detailed answer to each of their comments below.

1. Authors claimed the brightest single-photon emitter among TMD SPEs. It will be convincing if the authors include a table to compare the brightness from previous papers.

Response: We base this statement on the extremely low excitation powers used in our work, that nonetheless allowed to achieve high single photon generation rates reaching 69 MHz under excitation with 80 MHz, corresponding to 5.5 MHz in the first lens positioned above the sample. This compares favourably with, to the best of our knowledge, the only previous report of a single photon rate of 42 MHz in Luo et al *Nature Nanotechnology* 13, 1137–1142 (2018). Luo et al used a particle-on-a-mirror cavity, allowing redirection of a large fraction of emitted photons into the objective, thus reaching light collection efficiencies of 83% for SPEs, a feature which was not realised in our current sample, but with which our approach is potentially compatible. The highest photon count rate achieved by Luo et al was for excitation with a visible laser with 0.1 mW (at 78 MHz repetition rate). This is equivalent to ~pJ energy per pulse, i.e. around 1000 times higher pulse energy than we use. In our work we also compare SPEs on GaP antennas to SPEs produced by a more common method of placing WSe₂ on a low-index dielectric pillar as it was done in previous studies. We find that again this previously realised method (reported by Heriot Watt and Cambridge) results in SPEs with inferior properties, for example, requiring 100 to 10000 times higher laser powers for excitation.

Changes: We clarify this statement to explain that it refers to both the high single photon generation rate and the extremely low powers used to achieve that. We focus on comparison of our data with Luo et al. in more detail both in the main body of the text and in the conclusions.

2. SPEs due to strain

2-1. What was the yield to find single-photon emitters on GaP or SiO structures? It will be clear if a PL map is provided to visualize this, i.e. array of GaP (or SiO₂) structures with SPEs created in an array.

Response: We found that the SPEs are formed on nearly all nano-antennas with the yield above 90% out of more than 30 nano-antennas. This is consistent with the observations in low-index nano-structures of similar dimensions as reported in the works of Branny et al. & Palacios-Berraquero et al. (reference 6 and 7 in the main text). We did not measure large-area PL maps, but found SPE emission on every pillar, except for those where the monolayer was pierced.

Changes: We have edited the text to highlight the comparison with previous works.

2-2. For GaP dimer structure, where is the maximum tensile strain? Is it along the edge of the dimer structure? Is there a strain analysis on this structure? How likely is it that SPEs created at the hotspot?

Response: We extensively analysed the topography of strain induced in both single and double layer WSe₂ placed on top of dimer nano-antennas in our previous work Sortino, L. et al. *ACS Photonics* 7, 2413–2422 (2020) (reference 10 in the main text). There we demonstrate that the maximum of tensile strain is localized at the edges of the dimer nano-antennas, where we expect the formation of the SPEs. Our calculations show that for the nano-antennas with the radius above 200 nm, the edge and the gap regions exhibit similar photonic enhancement (see further discussion below in the answer to the first point raised by reviewer 2). We expect that the SPEs are formed at the edges of the nano-antennas where their position coincides with that of the photonic hotspots.

Changes: We have addressed the reviewer's concern in the main text by discussing more in detail the relationship between the strain distribution and the formation of the SPEs.

3. Polarization of SPEs

Was there a tendency in polarization from strain-induced SPEs? If so, are they preferably x-pol or y-pol?

Response: We provide a discussion on the polarization properties of SPEs in Supplementary Note III. As we show in Supplementary Figures 4 and 5, we did not observe any preferable polarisation for the strain-induced SPEs on the nano-antennas. However, we do observe the presence of different types of SPEs, either exhibiting the presence or absence of a fine structure splitting, as previously discussed in the literature [Kumar, S. et al. *Optica* 3, 882 (2016)].

Changes: We edited the text to address the lack of a preferential orientation in the SPEs polarization.

4. Longer lifetime from SPEs in GaP

An explanation for a longer lifetime with GaP dimer was briefly touched in the manuscript, however, it seems not easy to follow the logic by reading the manuscript. It would be great if authors can explain Figure 2(d) using the diagram shown in Figure 3(d) The lifetime of SPEs is dominated by faster decay between Γ_{nr} and Γ_r (as the authors also mentioned). So if Γ_r is accelerated, i.e. radiative decay is faster than non-radiative one, the radiated emission now become a major decay channel. However, in this case, the lifetime of an emitter should be shorter than the non-radiated decay lifetime. Therefore, with Γ_{nr} present, how can a lifetime get longer due to the Purcell effect?

Response: As Fig.2(d) shows, in our experiments the SPEs with particularly long lifetimes >20 ns are only found on GaP nano-antennas and all demonstrate high quantum efficiency therefore, using the diagram in Fig.3(d), we conclude that $\Gamma_{nr} \ll \Gamma_r$. Given that PL lifetimes around and exceeding 100 ns can be observed, also makes us suggest that some of the SPEs, which exhibit high quantum efficiency (i.e. $\Gamma_{nr} \ll \Gamma_r$) and lifetimes of few to tens of ns, may experience Purcell enhancement. We do not suggest that the Purcell enhancement may lead to longer PL lifetimes. The long PL lifetimes is an unusual feature of SPEs coupled to GaP nano-antennas that was discovered in this work.

Changes: We have carefully removed some possibly misleading statements in abstract and other parts of the manuscript, in order to clarify the above points.

5. Minor correction: blue data points throughout figures are not very visible (it looks either black or purple)

Response: We have changed the description of 'blue' to 'dark blue'. We hope that the red and dark blue data points have sufficient contrast.

Reviewer #2 (Remarks to the Author):

The manuscript by Sortino et al. reports on the coupling of single-photon emitters in WSe₂ to dielectric nanoantennas. The coupling increases the quantum efficiency from 4% to up to 86% and the PL brightness by a factor of 100 to 10000. What is new compared to other works, is the coupling to a *dielectric* resonator (compared to plasmonic structures in most other works), and thereby achieving a very high brightness of the single photon emission. The manuscript give a detailed insight into the physical origin and processes of the single photon emission in atomically thin WSe₂. The manuscript is well written, although I see some space for improvements (see comments), however, all points are only minor and should be easily addressable by the authors. It is technically sound and all conclusions and claims are supported by the data. The most difficult question is now, how significant are the results. Some of the results, e.g. the increased PL are nice, but I would not call it a major breakthrough. Others are completely new for this kind of single-photon emitter, e.g. the coherence measurements and the detailed analysis of the dynamics.

In conclusions, I think that this manuscript is an important step in understanding the SPE in 2D semiconductors. So, my feeling is, that it is important for the community, but maybe not of too much importance for the wider field.

Response: We thank the reviewer for their positive remarks and recognising the novelty of our work. We reply to their points below.

Comments/Minor issues:

1. Enhancement by the dimer antennas: In Fig. 1 a-e the properties of the GaP and SiO₂ structures are shown, and obviously, the dimer is far superior. However, the figure gives not all information directly.

a) Where is the maximum of the E-field enhancement exactly (at the outer edge or the edges forming the gap)? From the color plot, I believe the latter, while schematic drawing suggest the outer edge. This should be marked in the plots. [Later on, I found in the SI, that this could depend on the radius of the pillar, so the position changes in plot c-e?]

Response: The local maximum of the electric field intensity can be obtained at either the outer edges of the dimer or at the edges in the gap region. As we show in the image below, for a nano-antenna with a radius larger than 200 nm the hot-spots in the gap region and at the outer edges are similar in terms of field enhancement. We observed negligible differences in our simulations for the dipole emitters placed in the gap region or at the outer edges of the antenna. On the other hand, the SPEs are more likely to form at the positions of the highest strain at the outer edges of the nano-antennas, as shown in the schematic in Fig.1(a).

Figure R1: Electric field enhancement as a function of the position at the top surface of the nano-antenna ($z=200$ nm) under the plane wave excitation at 638 nm. Excitation is linearly polarized along the axis connecting the nano-pillar centres (the dashed line in the inset). The profiles are shifted vertically for clarity.

Changes: We updated the above figure in Supplementary Figure 1 and we have updated Supplementary Figure 2 to include the calculations of the radiation pattern for a dipole placed at the outer edges or the gap for a dimer nano-antenna with a radius of 200 nm.

b) Since it is unpredictable where the SPE are formed (or activated) and which polarization they will have, I would be good to show also the enhancement for different special positions and the perpendicular polarization on the nanoantennas.

Response: The role of the dipole position and orientation on the photoluminescence enhancement was extensively studied in our previous work (reference 9 in the main text). We provide below a figure of the simulations from our previous work, showing the radiative enhancement for different dipole position and orientation.

Figure R2: Radiative enhancement for a dipole emitting at 750 nm, placed at the top surface ($z=200$ nm) of a GaP nano-antenna (scale bar 50 nm). Adapted from L. Sortino et al. Nat. Comm. 2019 (reference 9 in the text).

Changes: We have updated the text to address the role of the SPE position relative to the antenna near-field, and that for larger nano-antennas the Purcell enhancement is less sensitive to the position of the SPE.

c) The antibunching dip is below 0.5, but not overwhelming. I don't see the reason why (the spectrum looks very clean). Can the authors discuss this. Do they find emitters with a better antibunching?

Response: The relatively shallow dip in the anti-bunching experiment is related to (1) the background produced by light scattered from the continuous wave (cw) laser at 730 nm and (2) the optical filters used in our experiment. We employed a pair of filters (long-pass 750 nm and short-pass 800 nm) creating a 50 nm transmission window for the light sent to the detector. As we show in the Supplementary Fig. 4, the presence of additional emitters at lower wavelengths, which are not fully filtered with this setup, may lead to unwanted photons to be sent to the photon-correlation detector. The light collection in the photon-correlation experiment was done using a multi-mode fibre, which further increased collection of unwanted laser photons. On the other hand, the PL spectrum in Figure 1f shows low background as the excitation is carried out with a pulsed laser at 638 nm and a single-mode fibre is used for detection.

Changes: We have updated the Methods section to provide the details for the photon correlation measurements.

2. When looking on the many complex fabrication steps of creating plasmonic or dielectric nanostructures, the additional dielectric spacer for plasmonic nanoantennas seems not a convincing argument of increased fabrication complexity (p2, l25). The reduced coupling and not lossless nature of the plasmonic part are convincing enough.

Changes: We have soften this claim in the main text. As the referee suggests, we highlight the detrimental effects in plasmonic structures such as proximity quenching and heating effects. We also point out that much higher excitation powers are required in SPEs coupled to plasmonic structures, most likely due to non-radiative losses.

3. Being a little bit pedantic, but the HBT setup was deceived by Robert Hanbury Brown and Richard Twiss, so the setup is called "Hanbury Brown and Twiss" or "Hanbury Brown-Twiss" setup. (At least without the dash between "Hanbury" and "Brown")

Response/Changes: We thank the reviewer for bringing this to our attention. We have corrected this appropriately.

4. When I first read the SI Note VI, I was confused what the optical losses by the "optical breadboard" are (my first idea: That's an aluminum plate with threads). Eventually I think the authors refer to the optical components on the optical breadboard, and that the efficiency is only 50% let me think there is a beam-splitter involved. But I would appreciate if the authors can clarify this, and add a schematic drawing of the whole setup. It would also be good to estimate the error here.

Response/Changes: We thank the reviewer for this comment. We have added an additional image in the Supplementary Note VI of the schematic of the experimental setup used in our experiments. As the review recognised, the main loss in this setup is a beam splitter used to direct light into the cryostat. We have corrected the table in SI Note VI to better address that the losses are due to the optical components installed on the optical breadboard. We changed the table where the losses were

listed. The optical losses previously referred to as the "Optical breadboard", are now listed as "Optical components (Cryostat)".

5. No issue but just a question out of curiosity: Why do the authors use a single mode fibre here, where they lose 98% of their photons. A 50um multi-mode fibre could easily collect 90% of all photons and would only slightly decrease the spatial resolution (which seems uncritical here).

Response: The use of a single mode fibre was chosen to improve the spatial resolution of our measurements. The use of a single mode fibre also helped to reduce the background from the laser light scattered by the nano-antennas which becomes non-negligible when collecting light from a larger area.

Reviewer #3 (Remarks to the Author):

The manuscript "Bright single photon emitters with enhanced quantum efficiency in a two dimensional semiconductor coupled with dielectric nano antennas" by L. Sortino et al., presents an interesting study on the incorporation of photonic enhancement through dielectric dimers that are used simultaneously to induce strain in WSe₂ flakes to create SPEs. The authors show a significant enhancement in photoluminescence brightness compared to pillars made of SiO₂. I think the findings of the manuscript are of interest to the generation and engineering of better quality single-photon emitters in 2D transition metal dichalcogenides. However, there is some ambiguity regarding the origins of such enhancement that I believe needs addressing.

Response: We thank the reviewer for their positive remarks on our work, we address their concerns below.

Due to the lack of any experimental quantification of the dimer's optical response, it doesn't seem easy to fully understand the origins of the enhancement reported by the authors. For example, how does the performance of the SPEs on GaP dimers compare to the random SPEs in a flake on a bare GaP substrate or that on a GaP pillar without increased field enhancement caused by the dimer? That would be very important to know and fully quantify since the brightness of TMD emitters has been reported to vary based on the substrate materials as well as encapsulation, through hBN, for example. Having this information would make claims about enhancements in brightness being due to Purcell enhancement more convincing.

Response: We trust that our work provides the most direct comparison reported so far between the technology used previously to realise SPEs on low-index (and low surface quality) dielectric nano-structures and the new technology developed in our work, based on high-index and high surface quality dielectric nano-structures. The differences that we observe are very clear and significant.

We did not observe SPEs of reliably measurable PL intensity on planar GaP. SPEs on planar SiO₂ have comparable properties to those formed on SiO₂ pillars, with the latter having the advantage of controlled positioning. We thus focussed on comparing the SPEs on GaP and SiO₂ pillars. We have not studied SPEs on GaP monomers (single pillars) but rather focussed on dimer nano-antennas, as these provide more pronounced and interesting photonic effects. For this reason, all our preliminary work (refs 9, 10 in the manuscript) was carried out on dimers. We can only comment here that we expect that GaP monomers may also be useful in improving the quantum efficiency and the lifetimes of SPEs. This may be a subject of another investigation beyond our current work.

Changes: We have edited the text of the manuscript to provide this information.

There is no much discussion in the paper about the quantification of the stress induced by the dimer and whether the structural properties of the dimer have been selected for optimized stress or for their optical response. Is the maximum stress expected to happen, as in the pillars, on the edges, or in the gap region where the field enhancement is maximum? This would play a role in the various calculations in the manuscript since the points of maximum optical enhancements don't necessarily coincide with the SPEs.

Response: We extensively analysed the strain topography induced in both single and double layer WSe₂ on top of dimer nano-antennas in our previous work Sortino, L. et al. *ACS Photonics* 7, 2413–2422 (2020) (reference 10 in the main text). There we demonstrate that the maximum of tensile strain is localized at the edges of the dimer nano-antennas, where we expect the formation of the SPEs. Our calculations show that for the nano-antennas with the radius above 200 nm, the edge and the gap regions exhibit similar photonic enhancement (see further discussion above in the answer to the first point raised by reviewer 2 and also in Figures R1 and R2). We expect that the SPEs are formed at the edges of the nano-antennas where their position coincides with that of the photonic hotspots.

I suggest including a better or zoomed up SEM image of the fabricated dimer as the one presented in Fig. S3 seems to show a low-quality structure. It would be insightful to characterize the scattering cross-sections of the actual dimers to confirm their performance matches that predicted from simulations.

Response: We agree with the reviewer that the study of the scattering cross-sections of the actual dimers are important. The study of the antenna resonances and their impact on the PL properties of two-dimensional WSe₂ has been the topic of our previous publication (reference 9 in the main text). Dimer nano-antennas with radius above 100 nm show broad resonances extending from 600 nm up to above 1000 nm, as we show in the Supplementary Figure 3. These optical resonances are much broader than the range in which SPEs are observed. Thus, the conclusions of the paper are not influenced by the exact form of the optical resonances of the dimers. Our dimer resonances and fabrication is also discussed in details in the work of Cambiasso, J. et al. *Nano Lett.* 17, 2017 (reference 8 in the main text).

Changes: We included a better SEM image of the fabricated dimer in the Supplementary Note 2.

What is the reason the localized SPEs spots on SiO₂ pillars can't be resolved in Fig. S7? Figure 1f suggests that the authors should be able to resolve the emitters. Also, what is the yield for SPE creation per GaP dimer? A similar PL map on dimers would be helpful to include.

Response: In Fig.S6 we show a map of PL intensity of WSe₂ at room temperature, showing that no enhancement of the PL is induced by the SiO₂ nano-pillars.

We found that the SPEs are formed on nearly all nano-antennas (yield 90%). This is consistent with the observations in low-index nano-structures of similar dimensions as reported in the works of Branny et al. & Palacios-Berraquero et al. (reference 6 and 7 in the main text). We did not measure large-area PL maps, but found SPE emission on every pillar, except for those where the monolayer was pierced.

Changes: We have now added this information in the manuscript.

Reviewers' Comments:

Reviewer #1:

Remarks to the Author:

All my concerns are satisfactorily addressed.

Reviewer #2:

Remarks to the Author:

All my concerns have been addressed in the response/the revised manuscript, and I can now recommend the publication of the manuscript.

Reviewer #3:

Remarks to the Author:

The authors have answered most of my questions. I am still curious on why they haven't observed any SPEs in flakes on bare GaP films, as reports have shown wrinkles, cracks or other sources of strain can cause SPEs to appear in bare flakes.

In general, I believe the manuscript is sufficiently novel and I recommend its publication.

Dear Editor,

We would like to resubmit our manuscript by Sortino L. et al. "Bright single-photon emitters with enhanced quantum efficiency in a two-dimensional semiconductor coupled with dielectric nanoantennas" for publication in Nature Communications.

We trust that we have addressed all the reviewers' comments, as we discuss below with a step-by-step response below.

REVIEWERS' COMMENTS

Reviewer #1 (Remarks to the Author):

All my concerns are satisfactorily addressed.

Response: We thank the reviewer for their positive evaluation of our efforts.

Reviewer #2 (Remarks to the Author):

All my concerns have been addressed in the response/the revised manuscript, and I can now recommend the publication of the manuscript.

Response: We thank the reviewer for recognition of our efforts and firmly supporting the publication of our work.

Reviewer #3 (Remarks to the Author):

The authors have answered most of my questions. I am still curious on why they haven't observed any SPEs in flakes on bare GaP films, as reports have shown wrinkles, cracks or other sources of strain can cause SPEs to appear in bare flakes.

In general, I believe the manuscript is sufficiently novel and I recommend its publication.

Response: We thank the reviewer for recognizing the novelty and supporting the publication of our work.

We believe the observation of SPEs in wrinkled WSe₂ on GaP is heavily affected by their low quantum efficiency and redirection of light in the high-index GaP substrate, resulting in a collection efficiency, extracted from our numerical simulations, of <2% in case of an in-plane dipole on a flat GaP substrate. In addition, the natural SPEs emission could be further quenched by the presence of a type II band alignment between WSe₂ and GaP and the requirement of high-power excitation, and thus higher efficiency of Auger annihilation. Both effects would reduce the excitonic population feeding into the SPE state and thus its light emission efficiency.